# An Update on RNA Virus Discovery: Current Challenges and Future Perspectives

**DOI:** 10.3390/v17070983

**Published:** 2025-07-15

**Authors:** Humberto Debat, Nicolas Bejerman

**Affiliations:** Instituto de Patología Vegetal, Centro de Investigaciones Agropecuarias, Instituto Nacional de Tecnología Agropecuaria (IPAVE-CIAP-INTA), Unidad de Fitopatología y Modelización Agrícola CONICET-INTA, Córdoba 5020, Argentina

**Keywords:** RNA virus discovery, viral evolution, metagenomics, zoonotic spillover, virus surveillance, emerging respiratory viruses, SARS-CoV-2 variants, pandemic preparedness

## Abstract

The relentless emergence of RNA viruses poses a perpetual threat to global public health, necessitating continuous efforts in surveillance, discovery, and understanding of these pathogens. This review provides a comprehensive update on recent advancements in RNA virus discovery, highlighting breakthroughs in technology and methodologies that have significantly enhanced our ability to identify novel viruses across diverse host organisms. We explore the expanding landscape of viral diversity, emphasizing the discovery of previously unknown viral families and the role of zoonotic transmissions in shaping the viral ecosystem. Additionally, we discuss the potential implications of RNA virus discovery on disease emergence and pandemic preparedness. Despite remarkable progress, current challenges in sample collection, data interpretation, and the characterization of newly identified viruses persist. Our ability to anticipate and respond to emerging respiratory threats relies on virus discovery as a cornerstone for understanding RNA virus evolution. We address these challenges and propose future directions for research, emphasizing the integration of multi-omic approaches, advanced computational tools, and international collaboration to overcome barriers in the field. This comprehensive overview aims to guide researchers, policymakers, and public health professionals in navigating the intricate landscape of RNA virus discovery, fostering a proactive and collaborative approach to anticipate and mitigate emerging viral threats.

## 1. Introduction

The perpetual threat posed by RNA viruses to global public health demands continual advancements in our ability to identify, characterize, and understand these dynamic pathogens. This review provides an updated perspective on the evolving landscape of RNA virus discovery, reflecting recent breakthroughs in technology and methodologies that have expanded our capacity to uncover novel viruses across diverse host organisms. The acceleration of viral discovery is not merely a reflection of improved detection methods but also a testament to the inherent complexity and adaptability of RNA viruses [1].

Over the past decade, the acceleration of virus discovery has been driven by a synergy between technological innovations in high-throughput sequencing, metagenomics, and bioinformatics, and a growing understanding of biological factors such as host–pathogen co-evolution and ecological dynamics, revolutionizing our approach to viral surveillance. These tools have enabled researchers to explore previously inaccessible niches, revealing an unprecedented diversity of RNA viruses [2]. The discovery of novel viral families and the identification of their reservoir hosts have illuminated the intricate relationships between viruses and their environments, with zoonotic transmissions playing a pivotal role in the dynamics of viral emergence [3].

This review probes into the expanding landscape of viral diversity, emphasizing the implications of RNA virus discovery for our understanding of disease emergence and the potential for pandemics. Despite the remarkable progress achieved, significant challenges persist in the field of viral discovery, ranging from limitations in sample collection to the complexities of data interpretation and the characterization of newly identified viruses. Addressing these challenges is crucial to bolstering our preparedness for emerging viral threats [4].

We explore the current state of knowledge regarding the factors influencing the dynamics of RNA viruses, with a particular focus on the intersection of viral ecology, host interactions, and the human–animal interface [5]. As we embark on an era of unprecedented viral discovery, it is imperative to critically assess the gaps in our understanding and chart a course for future research endeavors [6]. In this context, we propose a roadmap for advancing the field, advocating for the integration of multi-omic approaches, updated computational tools, and collaborative international efforts to navigate the intricate landscape of RNA virus discovery.

All in all, this review aims to provide a comprehensive introduction to the current state of RNA virus discovery, offering insights into the breakthroughs, challenges, and future perspectives that shape our ongoing efforts to unveil the viral dark matter landscape. Through this exploration, we strive to facilitate a proactive and collaborative approach that will enhance our ability to anticipate, understand, and ultimately mitigate the impact of emerging viral threats on global health.

## 2. Results

### 2.1. Technological Advancements in RNA Virus Discovery

The field of RNA virus discovery has undergone a revolutionary transformation, primarily driven by unparalleled technological advancements. High-throughput sequencing and metagenomic techniques have transcended traditional barriers, enabling researchers to venture into unexplored ecological landscapes with unprecedented precision [7]. These technological innovations have not only expedited the identification of known viruses but have also unearthed a plethora of previously undiscovered RNA viruses, expanding our understanding of the viral microcosm (Table 1).

The advent of next-generation sequencing platforms has ushered in an era of genomic exploration, facilitating the simultaneous analysis of vast viral populations within complex biological samples (Figure 1). Metagenomic approaches, coupled with advancements in sequencing technologies, empower researchers to conduct holistic surveys of diverse environments, from remote ecosystems to human clinical samples (Table 2). This synergy has resulted in the discovery of a myriad of RNA viruses, ranging from well-characterized pathogens to entirely novel viral entities [8,9,10,11]. Furthermore, portable sequencing platforms like Oxford Nanopore Technologies’ MinION have revolutionized field-based discovery due to their affordability and real-time capabilities. Concrete examples of its impact include its use in the rapid, culture-independent whole-genome sequencing of Nipah virus during outbreaks [12], its application in identifying recombinant enteroviruses responsible for outbreaks [13], and its utility in detecting novel plant viruses like macluraviruses and potyviruses in yam plants [14]. In addition, the application of these techniques has led to the discovery of novel RNA viruses in invertebrate hosts, such as ants, highlighting the continuous expansion of the RNA virosphere in understudied taxonomic groups [15]. Similarly, recent advancements in unbiased metagenomics have enabled the identification of entirely new lineages within the Orthornavirae kingdom, significantly broadening our understanding of RNA virus evolution and diversity across a wide range of hosts [16].

Recent breakthroughs in single-cell sequencing technologies offer a nuanced perspective on RNA virus discovery. By dissecting individual host cells, researchers can discern viral genomes with unprecedented resolution, unveiling the genetic diversity within infected populations [17]. For instance, single-cell sequencing has demonstrated its ability to detect viral transcripts and identify infected cell types in human skin biopsies for viruses like Merkel cell polyomavirus and human papillomaviruses [18]. It has also been applied to study the heterogeneity of influenza virus infections at the single-cell level, revealing how genetic variations contribute to diverse immune responses [19], and to identify novel host–virus interactions for giant viruses in marine environments [20]. This level of granularity proves invaluable in understanding viral quasispecies dynamics, host interactions, and the adaptive evolution of RNA viruses, providing a finer lens through which to explore the intricacies of viral biology.

### 2.2. Bioinformatics and Computational Tools

The deluge of sequencing data demands robust bioinformatics and computational tools for effective analysis. Advanced algorithms and machine learning models are instrumental in deciphering complex viral genomes, predicting potential host ranges, and discerning patterns of viral evolution [9]. Specific machine learning models such as deep learning models, random forests, and support vector machines (SVMs), alongside dedicated bioinformatics tools like VIRify, VirHostNet, and DeepViral [21], are instrumental in deciphering complex viral genomes, predicting potential host ranges, and discerning patterns of viral evolution. Beyond mere identification, AI/ML approaches are increasingly vital for functions such as predicting host tropism, identifying potential zoonotic reservoirs, and even inferring viral pathogenicity based on genomic features and protein structures (e.g., AlphaFold for protein modeling). The integration of these tools into RNA virus-discovery pipelines not only accelerates the identification process but also enhances our ability to extract meaningful insights from the vast genomic datasets generated. Notably, Serratus re-analyzed petabase-scale sequence data from public databases to facilitate the discovery of over 130,000 new RNA viruses, including several novel coronaviruses [22]. This innovative system was crafted to streamline ultra-high-throughput sequence alignment on a petabase scale. The researchers embarked on an exhaustive exploration, analyzing a vast dataset comprising 5.7 million biologically diverse samples, collectively amounting to 10.2 petabases. Focusing on the hallmark gene RNA-dependent RNA polymerase, this methodological approach yielded remarkable results, uncovering more than 10^5^ previously unknown RNA viruses. This significant discovery exponentially expanded the breadth of our understanding of viral species by roughly an order of magnitude, but more importantly, it provided a user friendly platform to foster and democratize virus discovery beyond the global north.

Beyond their application in data analysis, the emerging potential of AI agents is poised to transform the entire RNA virus discovery workflow through automation and acceleration. These intelligent systems can revolutionize the field by automating tedious data curation and preprocessing tasks, optimizing experimental design (e.g., through active learning for targeted sampling), and even generating novel hypotheses for viral origins or host interactions. For instance, AI-driven platforms can sift through vast public genomic datasets, identify subtle patterns indicative of novel viral families, and prioritize samples for further experimental validation, dramatically reducing the time from raw data to actionable insights [23]. This shift towards AI-powered automation promises to enhance efficiency, reduce human error, and enable researchers to explore previously intractable scales of data, thereby significantly accelerating the pace of RNA virus discovery and outbreak preparedness.

Technological progress in RNA virus discovery is further amplified by global collaborations and open data-sharing initiatives. International networks of researchers, epidemiologists, and public health organizations collaborate to pool resources, share methodologies, and collectively analyze data [24]. This collaborative approach accelerates the pace of discovery and ensures a comprehensive understanding of the global viral landscape.

### 2.3. Unraveling Novel Viral Families

The exploration of RNA virus diversity has led to the revelation of novel viral families, challenging existing paradigms and expanding the intricate tapestry of viral taxonomy [25]. Beyond the identification of individual viruses, the discovery of previously unknown viral families has redefined our understanding of the fundamental building blocks of the virosphere, emphasizing the dynamic nature of viral evolution and adaptation.

Recent breakthroughs in RNA virus discovery have necessitated a reevaluation of taxonomic frameworks. The identification of novel viral families poses challenges to traditional classification systems, prompting a paradigm shift in our conceptualization of viral diversity [26,27]. The taxonomic implications of these discoveries highlight a need for adaptive classification strategies that can accommodate the ever-expanding pool of newly identified viruses. The discovery of novel viral families not only broadens the spectrum of known viruses but also provides unique insights into the evolutionary dynamics of RNA viruses. Comparative genomics and phylogenetic analyses of these newfound families shed light on the origins, divergence, and adaptive strategies employed by diverse viral lineages [26]. Understanding their evolutionary trajectories is crucial for deciphering the mechanisms underpinning their emergence and persistence in various ecological niches.

### 2.4. Ecological and Zoonotic Dimensions of RNA Virus Discovery: Implications for One Health and Public Health Preparedness

The discovery of novel RNA viral families is continually expanding our understanding of the complex ecological relationships between viruses and their hosts [28]. These findings reveal not only the hidden breadth of viral diversity but also the subtle and dynamic interactions that viruses maintain within ecological networks. Each new viral lineage provides a unique lens through which to examine the evolutionary strategies of replication, transmission, and persistence, while also offering insights into the broader mechanisms shaping host–pathogen co-evolution.

Beyond their genomic novelty, these newly identified viral families underscore the functional diversity embedded within the virosphere [29,30]. Characterizing the replication strategies, structural features, and host interactions of these viruses is essential to understanding their ecological roles. Importantly, determining host range and tropism informs our ability to assess cross-species transmission potential and zoonotic risk—critical parameters for anticipating disease emergence.

The ecological significance of these discoveries extends well beyond virology, intersecting with key questions in evolutionary biology, biodiversity conservation, and ecosystem health. Viruses, as integral components of ecosystems, can influence population dynamics, species interactions, and even nutrient cycles. Zoonotic transmission is a central narrative in the evolutionary trajectory of many RNA viruses, including several of public health concern [31,32,33]. Zoonotic transmissions form a dynamic interface where the spheres of animal and human health intersect. As viruses navigate this boundary, they encounter diverse hosts, each presenting a unique set of challenges and opportunities for adaptation [34]. Identification of reservoir hosts is pivotal in understanding the dynamics of zoonotic transmissions [35]. Beyond the initial spillover event, there is a role of reservoir hosts in the amplification and maintenance of viruses within natural ecosystems.

Zoonotic transmissions are influenced by a complex interplay of evolutionary forces acting on both viruses and their hosts [36]. Evolutionary pressures, such as immune evasion and adaptation to new environments, shape the genetic makeup of viruses and influence their ability to cross species barriers [37].

Human activities and alterations in land use significantly influence the landscape of zoonotic transmissions [38,39,40]. There is a substantial impact of anthropogenic factors, including deforestation, urbanization, and agricultural practices, on the frequency and intensity of zoonotic events. Understanding these influences is crucial for predicting and mitigating the risk of viral spillover amidst environmental changes.

The evolving field of RNA virus discovery carries significant implications for our understanding of disease emergence and requires a fundamental change in global pandemic preparedness [41]. There is an imperative to adapt public health strategies to the evolving nature of RNA viruses and the associated challenges in mitigating the impact of potential pandemics.

The rapid pace of RNA virus discovery and the expanding catalog of viral diversity demand a fundamental shift in public health strategy, from reactive containment to proactive, adaptive response frameworks [42]. As newly discovered RNA viruses require adaptation in traditional approaches of surveillance, diagnostics, and intervention, public health systems must evolve to incorporate flexibility, foresight, and effective utilization of genomic data. Integrating viral discovery data into vaccine development pipelines, therapeutic design, and diagnostic assay development is essential to address the evolving nature of emerging and re-emerging pathogens.

Central to this adaptation is the adoption of data-driven decision-making processes. Real-time integration of genomic surveillance, predictive modeling, and risk mapping enables public health authorities to design more effective, context-specific interventions. Rather than relying solely on established clinical case definitions or historical pathogen profiles, public health preparedness can now incorporate insights from novel virus detection, viral evolution trends, and environmental triggers. The One Health approach, emphasizing the interconnectedness of human, animal, and environmental health, offers an essential framework for managing emerging infectious diseases [42]. Zoonotic spillovers rarely arise from isolated events; they are ecological phenomena shaped by biodiversity loss, land-use change, wildlife trade, and global mobility. As such, addressing them requires interdisciplinary collaboration across virology, veterinary medicine, ecology, epidemiology, and environmental science.

The global nature of emerging RNA virus threats also necessitates robust international cooperation [24,43]. Effective pandemic preparedness depends on the establishment and maintenance of cross-border networks for genomic surveillance, open-access data sharing, resource allocation, and coordinated response strategies. Platforms that support the timely sharing of viral sequences, epidemiological reports, and functional annotations can significantly reduce the delay between discovery and intervention. In this context, collaboration becomes both a scientific and geopolitical necessity. As our technical capacity to detect and interpret viral diversity expands, so too must our ethical frameworks. Ethical considerations surrounding RNA virus discovery include questions of informed consent, benefit sharing, and the responsible use of genetic information, particularly when research is conducted in biodiversity-rich but resource-limited regions [24]. Firstly, ethical data sharing is paramount and encompasses adherence to international agreements such as the Nagoya Protocol. This ensures that benefits arising from the utilization of genetic resources, including viral sequences, are shared fairly and equitably with the countries providing those resources. Secondly, the responsible use of viral genomic data originating from low-income countries requires careful consideration. This involves preventing exploitation, ensuring data security, and guaranteeing that research outcomes directly benefit the source communities and nations. Thirdly, issues pertaining to informed consent and ownership when working with indigenous communities are critical. Respect for tribal sovereignty, cultural sensitivities, and the right to self-determination must guide all interactions, ensuring free, prior, and informed consent for sample collection and data use, and recognizing community ownership over their biological resources and associated traditional knowledge. It is essential to ensure that the benefits of virus discovery are equitably distributed and that the rights of individuals and communities are respected. Balancing global health protection with justice, transparency, and respect for sovereignty is key to fostering trust and ensuring the long-term sustainability of collaborative research.

### 2.5. Persisting Challenges in RNA Virus Discovery

Despite remarkable strides in technology and methodology, the field of RNA virus discovery grapples with persistent challenges that demand innovative solutions and a concerted research focus [30]. This section delves into the enduring obstacles, ranging from sample collection intricacies to the complexities of viral characterization, and highlights the need for ongoing advancements to overcome these hurdles and propel the field forward.

#### 2.5.1. Sample Collection and Data Interpretation

The discovery of novel RNA viruses depends fundamentally on the quality and diversity of biological samples collected across ecological and clinical landscapes. However, obtaining representative and high-quality samples remains one of the most persistent challenges in the field [30]. Field sampling in remote or biodiverse habitats, collection from wildlife or vector populations, and handling of clinical specimens all involve logistical, technical, and biosafety constraints. Factors such as sample degradation, contamination, and limited access to cold-chain infrastructure frequently compromise data quality and hinder downstream molecular analyses. Inadequate sampling not only limits the resolution of viral diversity but also skews our understanding of host–virus associations and transmission dynamics.

These challenges are compounded by the ethical dimensions of sample acquisition, which are becoming increasingly salient as virus discovery expands into global and often vulnerable contexts [24]. Navigating the balance between scientific necessity and ethical responsibility requires meaningful engagement with local communities, respect for indigenous knowledge systems, and adherence to principles of benefit-sharing and informed consent. Particularly in regions rich in biodiversity and cultural heritage, virus discovery efforts must be guided by equitable frameworks that recognize the rights of stakeholders and promote long-term collaboration rather than extractive research practices.

Once samples are collected and sequenced, the complexity of data interpretation emerges as a second major bottleneck in the virus-discovery pipeline [7,24,30]. The high-throughput sequencing platforms used in metagenomic studies generate vast datasets, often containing millions of reads from mixed-origin nucleic acids. Distinguishing genuine viral sequences from background noise, laboratory contaminants, or host-derived elements demands rigorous quality control and the application of specialized bioinformatics pipelines. This is especially critical in the detection of low-abundance or highly divergent viral genomes, which may lack close homologs in existing reference databases.

Moreover, even when novel viral sequences are confidently identified, their biological relevance is not always clear. The challenge of interpreting functional significance, such as inferring host range, transmission potential, or pathogenicity, from genomic data alone remains unresolved in many cases. Advances in machine learning, k-mer–based classification, and domain-level annotation tools are beginning to bridge this gap, but substantial innovation is still needed to develop analytical frameworks capable of handling the scale and diversity of modern viromic datasets. Strengthening each of these pillars is essential to ensure that virus discovery continues to produce biologically meaningful, socially responsible, and actionable insights into the RNA virosphere.

#### 2.5.2. Viral Characterization Complexity

Identifying a novel RNA virus is only the first step in a much more complex endeavor: understanding its biology, host range, and potential impact on human and animal health. Comprehensive characterization of newly discovered viruses remains a significant challenge in the post-discovery phase [7,24,30]. Many viruses, particularly those identified through metagenomics, are known only from partial or complete genome sequences, with no corresponding cultured isolate or infectious clone. This lack of biological material limits experimental validation and hinders the study of viral replication dynamics, host cell interactions, tissue tropism, and pathogenicity. Even when isolation is possible, elucidating the full spectrum of host–virus interactions often requires the integration of diverse experimental systems, including animal models, cell culture, and in vitro binding assays, each with its own constraints and biases. Standardized protocols for viral characterization are inconsistently applied across laboratories, and the interpretation of findings can vary depending on the biological system and context. These discrepancies limit the comparability of results and pose barriers to building a coherent understanding of the virosphere.

Addressing these limitations requires more than technological advancement; it demands a truly interdisciplinary approach to viral discovery. RNA virus research sits at the intersection of virology, ecology, evolutionary biology, bioinformatics, structural biology, and public health [24]. However, fostering effective collaboration across these disciplines remains a nontrivial task. Differences in technical language, methodological assumptions, and research priorities can hinder communication and coordination between teams. Moreover, disparities in infrastructure and expertise, especially between institutions in high- and low-resource settings, exacerbate the difficulty of implementing shared standards and best practices.

One key area in need of improvement is data interoperability. As viral datasets grow in size and complexity, the need for standardized metadata, annotation protocols, and interoperable platforms becomes increasingly urgent. Harmonizing data formats and establishing community-driven guidelines for viral genome curation and characterization would enhance reproducibility and facilitate more meaningful comparisons across studies. Ultimately, the success of virus discovery efforts depends on our ability to bridge disciplinary divides and integrate knowledge at multiple levels, from molecular mechanisms to ecological context to public health relevance. Building collaborative frameworks that support equitable, interdisciplinary engagement will be essential to fully characterize the expanding RNA virosphere and translate discovery into actionable insight.

### 2.6. Future Directions and Collaborative Approaches

Anticipating the changing field of RNA virus discovery requires a proactive approach and collaborative frameworks across disciplinary and geographical boundaries [24]. Here we envision future directions for research, emphasizing the integration of advanced technologies and collaborative approaches to enhance our collective capacity to understand viral ecosystems and proactively address emerging threats (Table 3).

#### 2.6.1. Multi-Omic Integration and Predictive Modeling for Surveillance

A key direction for RNA virus discovery involves the convergence of technological innovation and systems-level biology. A significant area of focus is the integration of multi-omic approaches, genomics, metagenomics, transcriptomics, proteomics, metabolomics, and beyond, to achieve a comprehensive understanding of host–virus interactions and the ecological factors that shape viral emergence [29,44]. This multi-layered strategy enables the simultaneous analysis of viral genomic content and host cellular responses, offering insights into the molecular mechanisms of infection, immune evasion, replication strategies, and viral adaptation across diverse environments and species.

By combining these -omics layers, researchers can generate more robust hypotheses about virus function, host specificity, and transmission potential. Such integrative frameworks also facilitate the identification of conserved molecular signatures or host biomarkers associated with pathogenicity or zoonotic risk, critical features for surveillance, diagnostics, and therapeutic development. Complementing this multi-omic perspective is the growing need for powerful computational tools capable of processing and interpreting the increasing volume of data now produced by high-throughput platforms [9,22,30,45]. Future directions in bioinformatics emphasize the deployment of machine learning and artificial intelligence to support automated genome annotation, functional prediction, and taxonomic classification. Deep learning models trained on virome-scale datasets are increasingly used to identify viral sequences lacking homology to known taxa, anticipate mutational trajectories, and predict potential receptor usage, all without requiring prior experimental characterization.

Of particular importance is the role of predictive modeling in redefining virus discovery into a proactive tool for surveillance and pandemic prevention [9,31,46,47]. By integrating ecological, environmental, and epidemiological data, such as host distribution, land use, climate variability, and immunological pressures, computational models can forecast geographic regions and ecological interfaces where viral spillovers are most likely to occur. These predictive frameworks enable public health authorities to allocate resources more strategically, prioritize high-risk zones for targeted sampling, and implement early interventions before human outbreaks arise. As these tools mature, they hold the potential to not only accelerate discovery but to embed it within real-time, globally coordinated surveillance systems that can anticipate and mitigate the impact of emerging RNA viruses.

#### 2.6.2. Global Collaborations, Capacity Building and Data Sharing

As the discovery and surveillance of RNA viruses increasingly become global endeavors, the role of international collaboration and open data sharing is increasingly critical. The complexity and speed of viral emergence, particularly in the context of respiratory pathogens with pandemic potential, demand a coordinated, transnational response grounded in transparency, mutual support, and scientific exchange [24]. The future of RNA virus discovery envisions a interconnected research environment in which scientists, public health authorities, and institutions across sectors and regions work collaboratively to detect, understand, and contain viral threats.

Key to this approach is the development of standardized and interoperable data-sharing platforms that allow for the timely and secure exchange of genomic sequences, epidemiological metadata, functional annotations, and surveillance findings. Harmonized protocols for data curation, open-access repositories, and equitable sharing of benefits are essential not only for accelerating research but also for building trust among partners and enabling rapid, evidence-based responses to outbreaks. Removing administrative and political barriers to international data flow, while respecting privacy, sovereignty, and bioethical norms, is a necessary step in establishing a more cohesive global virology community. Equally vital to the future of RNA virus discovery is the deliberate and sustained investment in capacity building [24]. The burden of viral emergence is often highest in regions with limited surveillance infrastructure, fewer trained personnel, and restricted access to sequencing or computational resources. To address this disparity, global initiatives must prioritize training programs, fellowships, workshops, and regional hubs of excellence that equip researchers with the tools and knowledge needed to participate fully in discovery efforts. Capacity building is a central component of equitable and effective global health preparedness.

Fostering a diverse and globally distributed scientific workforce enhances the resilience and responsiveness of the global virology efforts. Local researchers are best positioned to detect and contextualize unusual viral signals within their ecological, cultural, and epidemiological settings. Supporting their inclusion not only enriches the quality of virus discovery but also empowers communities to manage their local health challenges and engage meaningfully with global science.

## 3. Discussion

### 3.1. The Importance of Virus Discovery in Understanding the Evolutionary Challenges of RNA Viruses

The evolutionary plasticity of RNA viruses underlies their capacity to adapt rapidly, evade host defenses, and generate novel variants with pandemic potential. Nowhere is this adaptability more evident than in the emergence of SARS-CoV-2 variants of concern, which have repeatedly reshaped the trajectory of the COVID-19 pandemic [48]. The continued appearance of immune-evasive or highly transmissible lineages underscores the fact that virus discovery is not merely a foundational exercise in cataloging biodiversity; it is a critical element of understanding and responding to the evolutionary challenges posed by RNA viruses [49].

Systematic virus discovery efforts provide the necessary genomic context to interpret evolutionary trajectories. By expanding the known diversity of viral sequences, particularly from under-sampled hosts and environments, we enhance the phylogenetic framework required to detect unusual divergence, recombination, or lineage expansion. This is particularly important for respiratory viruses, whose cross-species transmission events, often from birds, bats, or rodents, may occur long before human outbreaks are recognized. Surveillance-based discovery in animal reservoirs, environmental sources, and immunocompromised human hosts helps identify the evolutionary intermediates and precursors of emerging pathogens [50].

For example, the emergence of SARS-CoV-2 variants such as Alpha, Delta, and Omicron has been linked to prolonged intra-host evolution in immunosuppressed individuals, a process that likely accelerated the accumulation of mutations in key viral proteins like spike. Discovery pipelines that incorporate longitudinal sampling and deep sequencing in such patients offer rare windows into within-host evolutionary dynamics. Similarly, identifying novel betacoronaviruses and other RNA respiratory viruses in animal populations enables earlier risk assessment based on shared receptor usage, transmission potential, and genomic plasticity [51]. Moreover, virus discovery contributes directly to the identification of genomic features associated with adaptation, virulence, and immune escape. Novel RNA viruses characterized through metagenomics and structural prediction platforms can expose conserved or convergently evolving elements, such as RNA-dependent RNA polymerases, proteolytic cleavage sites, or glycosylation patterns on viral envelopes, that are central to viral fitness and host range shifts [52]. These insights not only inform the molecular basis of evolution but also refine targets for broad-spectrum therapeutics and vaccine design.

In the context of emerging respiratory diseases, the strategic integration of virus discovery into public health surveillance can act as an evolutionary early warning system. Discovering and monitoring RNA virus diversity across ecological interfaces allows for the identification of high-risk lineages before they acquire enhanced human transmissibility [53]. This anticipatory approach, grounded in evolutionary virology and empowered by discovery science, is essential for pandemic preparedness.

### 3.2. Zoonotic Interfaces and the Ecology of Viral Emergence

Zoonotic transmission remains one of the most critical and complex elements in the RNA virus emergence continuum [31,32]. As novel viruses are increasingly detected at the junction of wildlife, livestock, and human populations, it becomes clear that spillover events are not isolated anomalies but rather emergent properties of disrupted ecosystems. Land-use changes, such as deforestation and agricultural intensification, alongside global wildlife trade and urban encroachment, have profoundly altered host–pathogen dynamics. These disruptions facilitate the crossover of RNA viruses, many of which possess a high mutation rate and plastic genomes, enabling rapid adaptation to new hosts.

Yet, predicting which viruses may emerge remains a formidable challenge [31,54]. Reservoir host identification and ecological context are often poorly understood, especially in biodiverse regions where surveillance infrastructure is limited. Moreover, many spillovers are likely cryptic, involving asymptomatic or low-virulence infections that escape detection. To address these gaps, discovery efforts must go beyond cataloging viral genomes and incorporate ecological, behavioral, and evolutionary data to assess risk in a biologically meaningful way. Integrative approaches rooted in field ecology, environmental virology, and evolutionary modeling will be essential to anticipate and mitigate future zoonotic events.

### 3.3. Strategic Preparedness and the Ethics of Discovery

The accelerating pace of RNA virus discovery carries with it both promise and responsibility. While technological advances have dramatically expanded our capacity to detect novel viruses, their utility for global health depends on the translation of genomic data into actionable public health strategies [55]. Preparedness now requires adaptive, data-driven frameworks that can respond to the velocity and complexity of emerging threats. This includes the refinement of pathogen risk-ranking models, investment in pan-viral diagnostics and vaccines, and real-time integration of genomic surveillance into health systems.

However, alongside these strategic imperatives, ethical considerations must be foregrounded. Sampling in low-resource or biodiverse settings, often where novel viruses are most likely to be found, raises questions around informed consent, benefit-sharing, and the use of indigenous knowledge. Equally important is the governance of open-access data: while transparency is vital for rapid response, there must be safeguards to prevent misuse and ensure equitable participation in downstream research and innovation [24]. As RNA virus discovery becomes more globalized and technologically sophisticated, frameworks for ethical collaboration, data stewardship, and equitable capacity building will be central to ensuring that the benefits of discovery are shared and its risks responsibly managed.

### 3.4. The Next Ten Years of RNA Virus Discovery

The landscape of RNA virus discovery is expected to undergo significant evolution in the coming decade, driven by ongoing advancements and shifting research paradigms [24]. We stand positioned to delve deeper into enigmatic viral reservoirs, from the guano-laden roosts of bats to the hydrothermal vents of the deep sea, aiming to unveil ancestral strains and zoonotic hidden threats. The development of next generation of ultra-fast, ultra-affordable, and ultra-portable sequencers is anticipated to democratize virus detection, enabling real-time surveillance and rapid outbreak response that could lead to more immediate bedside diagnostics and on-the-spot environmental monitoring [56,57]. Sophisticated algorithms, empowered by artificial intelligence, are expected to enhance the deciphering of the avalanche of genetic data, unveiling the subtle markers of novel viruses and predicting their potential for zoonotic spillover or pathogenicity [9,31]. This shift in focus extends beyond readily classified viral families, venturing into the uncharted territory of “viral dark matter,” encompassing unculturable viruses, uncovering hidden reservoirs, and illuminating the intricate interface of viral interactions within their host organisms [58]. Finally, future efforts may enable the promise of personalized virus tracking, where individual genetic profiles inform tailored risk assessments and preventive measures, empowering a proactive approach to the ever-evolving viral landscape. In the coming decade, RNA virus discovery is projected to move beyond mere identification, evolving toward a comprehensive understanding of viral origins, evolution, and potential impact. This knowledge will forge the foundation for the development of intelligent tools for prevention, detection, and treatment, contributing to greater resilience in the face of the ever-shifting viral panorama.

## 4. Conclusions

### Shaping the Future of RNA Virus Discovery and Global Health Resilience

In conclusion, this review has examined the complex and changing field of RNA virus discovery, highlighting the significant impact of technological advancements, the increasing understanding of viral diversity, and the critical role of zoonotic transmission in shaping global health threats. Together, these insights emphasize the strong need for adaptive and forward-looking strategies in pandemic preparedness. As the field continues to evolve, meaningful collaboration emerges as an essential foundation, one that must span disciplinary, institutional, and geopolitical boundaries.

This review highlights a dynamic and interdependent relationship between viruses, hosts, and ecosystems, demanding integrated approaches that blend molecular insight with ecological and epidemiological context. The enduring challenges, ranging from sampling and data interpretation to viral characterization and equitable access, call for a sustained, coordinated effort across the scientific community. RNA virus discovery will increasingly rely on the integration of advanced innovation, inclusive global collaboration, and predictive capacity.

As we move deeper into less-explored viral ecosystems, our commitment to ethical practice, scientific integrity, and community engagement must remain unwavering. Fostering a collaborative and equitable research culture is not only ethically important, it is essential for building global resilience to future pandemics. The roadmap outlined in this review serves to highlight the importance of action for researchers, public health leaders, and policymakers to co-create a more prepared, responsive, and inclusive framework for RNA virus discovery, one that meets the challenges of today and addresses those of tomorrow.

## Figures and Tables

**Figure 1 viruses-17-00983-f001:**
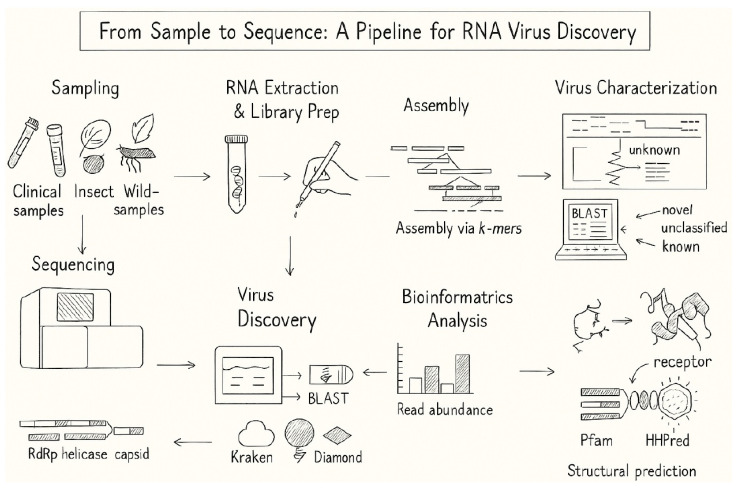
A schematic representation outlining the key stages in the contemporary workflow for RNA virus discovery. The process begins with diverse sampling from clinical, environmental, vector, or wildlife sources to capture a broad range of viral reservoirs, followed by RNA extraction and library preparation, which includes RNA fragmentation, adapter ligation, and quality control to ensure high-quality input for sequencing and post processing (e.g., FastQC, Trimmomatic). In the sequencing and assembly phase, raw reads generated from high-throughput platforms are processed, for instance, using k-mer–based approaches (e.g., SPAdes, MEGAHIT) to reconstruct viral genomes. The resulting viral contigs then enter the bioinformatics analysis pipeline, where viral sequences are annotated, quantified, and compared against reference databases using tools such as BLAST, Diamond, Kraken, LucaProt or Serratus to identify known viral elements and potential novelties. Phylogenetic placement (e.g., IQ-TREE, RAxML) enables preliminary classification into known, unclassified, or novel viral groups. Virus characterization incorporates functional and structural annotation through domain prediction (e.g., RdRp, helicase, capsid), protein structure modeling (e.g., using AlphaFold or HHpred), and receptor-binding inference to understand viral biology and potential pathogenicity. This multi-step workflow illustrates the integration of field sampling, molecular biology, high-throughput sequencing, and computational tools required to advance RNA virus discovery in the post-genomic era.

**Table 1 viruses-17-00983-t001:** Timeline of key technological advancements in RNA virus discovery.

Period/Year	Milestone/Advancement
** *Early 2000s* **	**Foundational Sequencing and Bioinformatics** Completion of the Human Genome Project, establishing large-scale sequencing capabilities. Widespread adoption of Sanger sequencing and initial development of foundational bioinformatics tools.
** *Mid 2000s* **	**First-Generation Next-Generation Sequencing (NGS)** Introduction of high-throughput sequencing platforms (e.g., 454 Life Sciences, Illumina Genome Analyzer), enabling parallel sequencing and significantly reducing costs and time per base.
** *Late 2000s–Early 2010s* **	**Metagenomics and Metatranscriptomics Emergence** Development and widespread application of unbiased metagenomic and metatranscriptomic approaches, allowing for the discovery of novel viruses without prior cultivation.
** *Mid 2010s* **	**Third-Generation Sequencing and Portability** Commercialization of long-read sequencing technologies (e.g., Pacific Biosciences, Oxford Nanopore Technologies’ MinION), offering real-time data, portability, and improved resolution for complex genomes.
** *Late 2010s–Early 2020s* **	**Advanced Computational and Single-Cell Approaches** Significant advancements in single-cell sequencing, enabling analysis of viral presence and gene expression at individual host cell resolution. Increased integration of Artificial Intelligence (AI) and Machine Learning (ML) for tasks like viral host prediction and classification. Development of large-scale cloud-based bioinformatics infrastructures (e.g., Serratus) for petabase-scale data analysis.
** *Recent/Ongoing* **	**Integrated Omics and Global Collaboration** Expansion of multi-omics integration (genomics, transcriptomics, proteomics) for a holistic view of virus–host interactions. Growth of global data-sharing initiatives and collaborative networks accelerating virus discovery and surveillance efforts.

**Table 2 viruses-17-00983-t002:** Key challenges and future perspectives in RNA virus discovery.

Challenges	Perspectives
**Sample Collection** Remote access, degradation, contamination, and lack of standardized methods limit the quality and scope of samples. (See Section 2.5.1.)	**Multi-Omics Integration** Combining genomics, transcriptomics, and proteomics to illuminate virus–host dynamics. (See Section 2.4.)
**Data Overload and Interpretation** Discriminating real viral sequences from noise in large metagenomic datasets remains difficult. (See Section 2.5.2.)	**AI-Powered Discovery** Machine learning models to enhance virus classification, host prediction, and outbreak risk assessment. (See Section 3.4.)
**Viral Characterization** Functional and biological validation lags behind genomic identification due to lack of isolates and models. (See Section 2.5.2.)	**Portable Sequencing Platforms** On-site and real-time virus detection through ultra-portable, affordable sequencing technologies. (See Section 2.1.)
**Taxonomic Uncertainty** Novel lineages challenge existing classification schemes, demanding more flexible and dynamic frameworks. (See Section 2.5.2.)	**One Health and Ecological Frameworks** Integrated views of human, animal, and environmental health to contextualize virus emergence. (See Section 2.4.)
**Ethical and Legal Issues** Sample ownership, informed consent, and fair benefit sharing are unresolved, especially in biodiverse regions. (See Section 3.3.)	**Global Equity and Capacity Building** International collaboration, open-access data, and inclusive training to democratize discovery. (See Section 3.3.)

**Table 3 viruses-17-00983-t003:** Emerging tools and innovations shaping the future of RNA virus discovery.

Innovation	Impact On Virus Discovery
** *Petabase-Scale Alignment* **	Enables detection of highly divergent RNA viruses from massive sequencing repositories using signature viral markers like RdRp.
** *Single-Cell Sequencing* **	Provides resolution at the level of individual infected cells, uncovering within-host diversity and viral replication dynamics.
** *Ultra-Portable Sequencers (E.G., Minion)* **	Facilitates real-time, field-based virus detection for outbreak response and surveillance in remote locations.
** *Predictive Spillover Modeling* **	Integrates ecological, host, and evolutionary data to identify high-risk interfaces for zoonotic emergence.
** *Ai-Based Genome Annotation* **	Automates and accelerates the functional classification of viral genomes, improving throughput and reliability.
** *Viral Dark Matter Exploration* **	Focuses on uncovering unclassified or unculturable viruses, expanding the known virosphere and redefining taxonomy.
** *Cloud-Based Discovery Pipelines* **	Democratizes computational power, enabling global researchers to analyze viral metagenomic data at scale.
** *Synthetic Viromics* **	Allows for the synthetic reconstruction and functional testing of candidate viruses to evaluate host range and pathogenicity.

## Data Availability

Not applicable for this review.

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
