# Peer review of "An Update on RNA Virus Discovery: Current Challenges and Future Perspectives"

_viruses, 2025, doi:10.3390/v17070983_

Round 1
Reviewer 1 Report
Comments and Suggestions for Authors
Debat and Bejerman present a comprehensive review on RNA virus discovery, highlighting recent advancements, persisting challenges, and future directions. It synthesizes a significant breadth of literature and includes forward-looking reflections on virus surveillance infrastructure, ethics, and the integration of multi-omic approaches. The manuscript is timely and relevant, especially in the context of post-pandemic preparedness, and has the potential to be a widely cited reference in the virology community. However, it also requires focused revisions to improve its clarity, remove some redundancy, and deepen its technical utility for the intended audience, before it is fully considered for publication.
Structure Comments
Overuse of Generalizations and Buzzwords
The text has excessive use of generalized phrases (e.g., "transformative decade," "unprecedented resolution," "viral dark matter") that may weaken the precision expected of a technical review. Trim such language throughout, especially in sections like 2.6, 2.9, and the Conclusion.
Redundancy in Sections 2.4–2.7
There is a lot of conceptual overlap across sections covering ecological significance, zoonotic spillover, One Health, and public health preparedness. These could be synthesized more tightly or even merged. For instance, the arguments made about the ecological roles of viruses and zoonotic risk are reiterated in multiple forms across several sections.
Figure 1: Workflow Utility
The description of Figure 1 is detailed, but the figure itself could benefit from color-coded stages or an expanded legend. Additionally, including real software examples per step (e.g., for de novo assembly, quality control, or phylogenetic reconstruction) would increase its practical value.
Box 1 and Box 2
These boxes are informative but inconsistent in style. Box 1 presents both “Challenges” and “Perspectives” but lacks clear references to later sections where those challenges are unpacked. Consider explicitly linking these to corresponding subsections (e.g., "See 2.8.2 for challenges in viral characterization").
Missing Case Examples for Methods
You describe Serratus, single-cell sequencing, and portable platforms, but the text would benefit from including a few concrete examples of how these have contributed to recent discoveries. For instance, mention which new virus or viral family was first reported via Serratus or MinION to anchor these methods in real discoveries.
Ethical Discussion Needs More Precision
The ethical framework discussion (lines ~238–244 and again in 3.3) is thoughtful but could be strengthened by distinguishing between (i) ethical data sharing (Nagoya Protocol context), (ii) responsible use of data from low-income countries, and (iii) consent/ownership in indigenous communities. These are different concerns and deserve to be treated with nuance.
Minor Comments
L9–10: "elusive pathogens" could be toned down—scientifically, they're not elusive if metagenomics/metatranscriptomics can find them.
L33–36: Consider rewriting to better separate the role of technology vs. biological factors in accelerating discovery.
L66: “Revolutionary transformation” is a bit of an overstatement; “significant progress” might suffice.
L105–107: Name specific machine learning tools/models used for host prediction or virus classification.
L214–216: Distinguish between diagnostics for known viruses vs. prediction of spillover for unknown viruses.
L291–299: When describing limitations in characterizing newly discovered viruses, mention that viral culture systems and reverse genetics platforms are still inaccessible for most novel viruses.
L481–502: Section titled “The next ten years…” reads more like a vision essay. That’s fine, but it needs clearer framing to match the tone of the earlier sections.
Reviewer 2 Report
Comments and Suggestions for Authors
In this comprehensive review, Debat and Bejerman present an insightful and up-to-date overview of the landscape of RNA virus discovery. The authors systematically detail recent technological advancements—particularly in high-throughput sequencing, metagenomics, and computational tools—that have significantly expanded our understanding of viral diversity. They highlight the discovery of novel viral families, discuss the ecological and zoonotic implications of RNA viruses, and emphasize the importance of a One Health framework for surveillance and preparedness. The review also provides a thoughtful roadmap for future research, advocating for multi-omic integration, international collaboration, and ethical considerations in global virus discovery efforts. Overall, this manuscript makes a valuable contribution to the field and will serve as a useful resource for virologists, epidemiologists, and public health professionals. I have a few minor suggestions for the authors to consider:
- In Section 2.1, the authors provide a concise summary of technological advancements in RNA virus discovery. To enhance the readability and contextual understanding of this section, I recommend including a timeline that highlights significant milestones in the development and application of these technologies.
- In Section 2.2, the authors discuss bioinformatics and computational tools relevant to RNA virus discovery. I suggest expanding this section with a more detailed discussion of specific advanced bioinformatics tools and AI/ML models that have been applied in this field, accompanied by additional references. Furthermore, the authors may consider addressing the emerging potential of AI agents in automating and accelerating RNA virus discovery workflows, which could have transformative impacts on future research.
Reviewer 3 Report
Comments and Suggestions for Authors
The Review article by Debat and Bejerman is a timely and comprehensive assessment of the current status of RNA virus discovery including challenges, accomplishments, and future directions/needs. The article is well-written and covers all of the key areas of interest to a wide audience. I have two suggestions to help strengthen the article:
- Although the authors include the specific example of SARS-CoV2 in the Discussion section, the article would benefit significantly if the authors also included several specific examples of viruses or techniques throughout the Results section, rather than simply using broad and general language throughout the Results sections.
- Both Box 1 and Box 2 should be reorganized into more sophisticated Tables (perhaps with color) that would easier for the reader both visually and comprehensively. As is, the Boxes/Tables are rather pedestrian.
